# Rotating Lorentz Force Magnetic Bearings’ Dynamics Modeling and Adaptive Controller Design

**DOI:** 10.3390/s23208543

**Published:** 2023-10-18

**Authors:** Feiyu Chen, Weijie Wang, Shengjun Wang

**Affiliations:** 1Graduate School, Space Engineering University, Beijing 101400, China; 16638396564@163.com; 2Department of Aerospace Science and Technology, Space Engineering University, Beijing 101400, China; wangshengjun2@sina.com

**Keywords:** rotating Lorentz force magnetic bearings, adaptive control, multiple closed-loop control, point to stability

## Abstract

To address the issues of our agile satellites’ poor attitude maneuverability, low pointing stability, and pointing inaccuracy, this paper proposes a new type of stabilized platform based on seven-degree-of-freedom Lorentz force magnetic levitation. Furthermore, in this study, we designed an adaptive controller based on the RBF neural network for the rotating magnetic bearing, which can improve the pointing accuracy of satellite loads. To begin, the advanced features of the new platform are described in comparison with the traditional electromechanical platform, and the structural characteristics and working principle of the platform are clarified. The significance of rotating magnetic bearings in improving load pointing accuracy is also clarified, and its rotor dynamics model is established to provide the input and output equations. The adaptive controller based on the RBF neural network is designed for the needs of high accuracy of the load pointing, high stability, and strong robustness of the system, and the current feedback inner loop is added to improve the system stiffness and rapidity. The final simulation results show that, when compared to the PID controller and robust sliding mode controller, the controller’s pointing accuracy and anti-interference ability are greatly improved, and the system robustness is strong, which can effectively improve the pointing accuracy and pointing stability of the satellite/payload, as well as provide a powerful means of solving related problems in the fields of laser communication, high score detection, and so on.

## 1. Introduction

Presently, space missions such as laser communication and high-score reconnaissance have increasingly high demands for rapid maneuverability and pointing accuracy of the payload. Both the satellite platforms are required to have ultra-stable pointing stability, ultra-sensitive attitude maneuverability, and ultra-precise load pointing accuracy [1,2,3]. In addition, compared to the world’s top level, China’s agile satellite attitude maneuver angular acceleration is small, and the accuracy and stability of the output torque are poor, resulting in the poor pointing accuracy and pointing stability of the load and a poor maneuvering capability.

In order to solve the above problems, there are currently three solution ideas: one is the integrated design of the satellite platform and the load, which can effectively avoid the high-frequency vibration and high-order perturbation brought about by fuel shaking, the vibration of flexural parts, and measurement and control asynchrony. However, because the specialized nature is strong, the cost of a single star is high, which is not conducive to multi-satellite network work. The second is to develop high-precision control actuators, but the research and development cycle is long, which is not conducive to accelerating the use of space. The third is to use the stabilized platform to isolate the vibration transmission between the satellite platform and the payload, optimize the control algorithms, and improve the stabilized platform’s ability to control the payload’s maneuvering and pointing. Therefore, under the current technical conditions, the use of stabilized platforms has more practical value for improving the performance of agile satellites [4,5,6,7,8,9].

Stabilized platforms can be divided into mechanically stabilized platforms, mechanically and electromagnetically stabilized platforms, and electromagnetically stabilized platforms according to the difference in the force between the axis systems. Traditional mechanical stabilized platforms have friction moments between the axes, which affects the pointing accuracy and service life of the platform. Meanwhile, the magnetic levitation bearing has the characteristics of being frictionless, having a long life, having no displacement negative stiffness, and being actively controlled, which are why it is widely used. Therefore, the current mainstream stabilized platforms are mechanical electromagnetic platforms and electromagnetic stabilized platforms. The core idea of the mechanical electromagnetic platform is to use the magnetic levitation bearing to control the important degrees of freedom to ensure the accuracy, and use the mechanical bearing to ensure the stability, of the remaining degrees of freedom. This kind of idea is common in agile satellites: The French SPOT series of Earth observation satellites use the magnetic levitation flywheel as the attitude control system to achieve high-resolution Earth observation [10]. Fang Jiancheng’s team at the Beijing University of Aeronautics and Astronautics (BUAA) has realized the high-precision attitude control of large remote-sensing satellites based on the magnetic levitation control moment gyroscope developed [11,12,13]. Ren Yuan’s team at the University of Aerospace Engineering proposes a magnetic levitation control sensitive gyro that integrates moment output and attitude measurement to achieve attitude measurement and control of the deflected two degrees of freedom. In summary, the idea of the mechanical electromagnetic platform is to place one or more magnetic levitation components in the corresponding degrees of freedom of the satellite as actuators or measuring devices, to control the stable pointing and agile maneuvering of the satellite/payload. However, because the magnetic levitation only controls part of the degrees of freedom, it is easy to produce high-frequency micro-amplitude vibrations in the process of momentum exchange, which restricts the pointing accuracy of the payload. Moreover, the control bandwidth is narrow, so it cannot deal with high-frequency and high-order perturbations, and it is necessary to additionally increase the band-pass filter, which, in turn, increases the complexity of the system [14,15,16,17].

Electromagnetic stabilized platforms can well-solve the problems existing in mechanical electromagnetic platforms. As the full degrees of freedom of the load/platform are supported by magnetic levitation bearings, it is frictionless, requires no lubrication, and can be actively controlled, which can avoid friction torque and vibration transmission. The magnetic levitation rotary joint designed by the Beijing Institute of Control Engineering uses a magnetic levitation bearing to support a single-degree-of-freedom swing of the load, which can solve the imaging accuracy problem of the variable speed phase in the attitude maneuver of remote-sensing satellites [18]. In addition, the magnetic levitation airborne inertial stabilization platform designed by Liu Kun’s team from the University of National Defense Technology supports a two-degree-of-freedom deflection, which realizes the high-resolution airborne remote sensing of the Earth [19,20]. The delicate high-frequency test satellite platform developed by the Shanghai Academy of Space Technology supports two-degree-of-freedom deflection and two-degree-of-freedom, which has the “double super” capability of an ultra-high pointing accuracy and ultra-high stability [21,22]. However, the magnetic levitation stabilization platforms developed by the above units either use magnetoresistive magnetic bearings, whose nonlinear and strong coupling characteristics easily lead to low control accuracy, or they only have individual degrees of freedom movability and the rest of the degrees are freedom fixed.

The new seven-degree-of-freedom Lorentz force magnetic levitation stabilization platform proposed in this paper adds a single degree of freedom for overall deflection on top of the six-degree-of-freedom motion of the load module, which can more effectively isolate the vibration transmission and high-frequency perturbation of satellites, and adopts the Lorentz force magnetic bearings in all the seven degrees of freedom, which ensures that the input current is linear with the output force/torque, and improves the pointing precision and pointing stability of the loads. The seven-degree-of-freedom novel Lorentz force stabilized platform consists of rotary magnetic bearings, linear magnetic bearings, and orthogonal magnetic bearings, in which the linear magnetic bearings and orthogonal magnetic bearings jointly control the two-degree-of-freedom deflection and three-degree-of-freedom translation to isolate the high-frequency vibration and higher-order perturbation between the load and the satellite; the rotary magnetic bearings act as the angular displacement actuators to improve the angular displacement, and they drive the load to achieve the fast maneuver and stable pointing.

Therefore, the controller design of the rotary magnetic bearing in this paper is directly related to the pointing performance of the load. At present, the controllers of spacecraft actuators generally have PID controllers, robust sliding mode controllers, feedforward decoupling controllers, controllers based on perturbation state observers, and so on. Among them, PID controllers and feedforward decoupling controllers require accurate mathematical models and parameters, while the higher-order perturbations in space are difficult to be modeled accurately; robust sliding mode controllers need to give upper and lower bounds of the perturbations and improve the robustness of the system under the premise of sacrificing the accuracy; and the control method based on the state observer of the disturbance, although it can accurately estimate the perturbations and isolate the perturbations by the accompanying filters, will increase the complexity of the system greatly. Therefore, in this paper, we present our design for an adaptive controller based on the RBF neural network for the structural characteristics and working principle of rotating magnetic bearings. This design meets the requirements of achieving fast attitude maneuver of the load, stabilizing load attitude, and improving pointing accuracy. We verify the effectiveness of the method with the results of modeling simulation.

## 2. Design of Lorentz Force Universal Magnetic Levitation Configuration 

### 2.1. Seven-Degree-of-Freedom Universal Suspension Scheme

The Lorentz force magnetic levitation universal stability platform is a seven-degree-of-freedom magnetic levitation mechanism made up of a load tank, a frame cabin, and a platform cabin, as well as two pairs of rotating Lorentz force magnetic bearings and pairs of linear and orthogonal Lorentz force bearings. The right-hand rule is used to establish the direction of the Z-axis, and the linear Lorentz force magnetic bearing controls the translational movement of the load chamber in the Y direction, as shown in Figure 1. Assuming the geometric center of the load tank serves as the origin for the coordinate system, the geometric centers of the load tank and the frame cabin are coincident. The active rotation axes of the load tank and the frame cabin are the Y- and X-axes, respectively, and the direction of the orthogonal Lorentz force magnetic bearing controls the translation and rotation of the load chamber in the X and Z directions. The payload tank is finally suspended in five degrees of freedom. To achieve a single-degree-of-freedom deflection, two sets of rotating Lorentz force magnetic bearings control the load chamber and frame compartment around their respective rotating axes. Among them, the inner RLFMB directs the rotation of the load compartment in the Y direction, while the outer RLFMB directs the rotation of the frame compartment in the X direction. Together, the two are able to universally deflect the load. The general layout of the universal stable platform for the Lorentz force is depicted in Figure 2.

As can be seen in Figure 3, the load chamber’s three degrees of freedom of X, Y, and Z translational magnetic levitation and rotation are controlled by orthogonal, linear Lorentz force magnetic bearings. Figure 1 shows that there is little space between the load tank and the frame compartment, which means that the load tank rotates in the same direction as the outside RLFMB control load compartment in the X and Z directions. The inner and outer RLFMB control how the load chamber and the frame compartment rotate along the Y- and Z-axes, and furthermore, ensure the maneuvering coverage and pointing stability of the load carried by the load compartment to the 1π space. As a result, the five-degree-of-freedom movement controlled by orthogonal and linear magnetic bearings serves to keep the load chamber in the frame cabin with good magnetic levitation performance.

To accomplish a steady direction of movement of the load to the target, the Lorentz force magnetic levitation universal stabilization platform modifies the operating current of each magnetic bearing to manage its output force and angular moment. The computer calculates the magnitude of the pointing error while the target is in the load field of vision, and then it modifies the working current of each magnetic bearing to accomplish closed-loop control.

### 2.2. RLFMB Structure and Principle

#### 2.2.1. RLFMB Structural Design

Figure 4 depicts the RLFMB’s structural layout. The RLFMB’s stator is made up of an exterior magnetic conducting ring, an outside magnetic steel, an inner magnetic steel, and an inner magnetic conductive ring, which are arranged radially from outside to inside. The rotor is made up of a coil bracket and a coil. The two groups of radially symmetrical inner and outer magnets have the same magnetization direction, and the outer magnet and the inner magnet are both permanent magnets that are radially magnetized. For the magnetic circuit to be closed, the outer and inner magnetic conductive rings are made of magnetically conductive materials. The coil bracket considers the good thermal conductivity of metal materials while choosing materials, but it will generate induced electromotive force to impede the rotor rotation. Non-metallic materials frequently result in rotor output attenuation at high temperatures because of their poor heat conductivity. Because RLFMB achieves stable pointing, deflection swing movement, no continuous rotation, and less heat generation, the coil bracket can be made of polyimide, a non-metallic material. The coil is made of copper wire.

To ensure that RLFMB does not maneuver uncontrollably at a great angle of the load compartment/frame compartment due to instability, the inner and outer magnets of the machine are left open under this structure. This also restricts the direction of the load to a specific range, creating a safe working environment. The magnet’s axial thickness can be raised while maintaining the same mass due to the structure’s simultaneous reduction in the magnet’s radial size, which ensures that the magnetic circuit is closed and minimizes magnetic leakage.

#### 2.2.2. Analysis of the RLFMB Principle

Because the two groups of inner and outer magnets are symmetrically charged in the same direction along the radial axis and the coil current direction is opposite, a pair of equally large reverse forces are produced, generating torque and causing the coil bracket and the coil to deflect in the gap between the inner and outer magnets when the current is passed through the coil. To create a fixed shaft rotation with a single degree of freedom, the coil bracket is connected through an output shaft to the load compartment/frame compartment. The RLFMB output moment diagram is displayed in Figure 5.

## 3. RLFMB Rotational Dynamics Modeling

### 3.1. RLFMB Equivalent Magnetic Circuit Model

The inner and outer magnets in the RLFMB magnetic circuit provide the magnetomotive force. Along the radial direction, from the magnetic resistance, the outer magnetic conductive ring, outer magnetic steel, working air gap, inner magnetic steel, and inner magnetic conductive ring follow. The inner and outer magnetic rings are joined as a single unit, and the inner and outside magnetic steel form an open sector-shaped magnet. The magnetic induction intensity at the working air gap are oriented in the same direction for the two groups of inner and outer magnets because they are radially symmetrical. Figure 6 depicts the analogous magnetic circuit of one side of the RLFMB, and radial symmetry on the other side is the same.

Ignoring the influence of magnetic circuit leakage magnetism and the magnetic field generated by the coil, according to the equivalent magnetic circuit method, its magnetic flux can be expressed as follows:(1)$$\begin{array}{l} \phi =\frac{\sum_{s}F_{S}}{\sum_{s}R_{S}+\sum_{i=1}^{4}R_{i}}\\ s\in (O-,O+,I-,I+) \end{array}$$

In the above formula, $F_{O-},F_{O+},F_{I-},F_{I+}$ is the magnetomotive force of four permanent magnets, $R_{1},R_{2}$ is working air gap magnetoresistance, $R_{3},R_{4}$ is the magnetoresistance of the inner and outer magnetic rings, $R_{O-},R_{O+},R_{I-},R_{I+}$ is the magnetoresistance of four permanent magnets, $H_{c}$ is the coercive force of radially magnetized permanent magnets, and ϕ is the magnetic flux at the operating air gap.

The magnetic induction intensity is expressed as follows:(2)B=ϕ/S

This can be brought into Equation (1):(3)$$B=\frac{H_{c}(2l_{p1}+2l_{p2})}{(\frac{l_{1}}{S_{1}\mu_{0}\mu_{r}}+\frac{l_{2}}{S_{2}\mu_{0}\mu_{r}}+\frac{2\delta }{\mu_{0}S}+\frac{2l_{p1}}{S_{p1}\mu_{0}\mu_{r}}+\frac{2l_{p2}}{S_{p2}\mu_{0}\mu_{r}})S}$$

In the above formula, $l_{p1}$ and $l_{p2}$ are the magnetization lengths of the inner and outer permanent magnets, $l_{1}$ and $l_{2}$ are the thicknesses of the inner and outer permanent magnets, $S_{1}$ and $S_{2}$ are the circumferential surface areas of the magnetic conductive ring surrounding the inner and outer permanent magnets, δ is the length of the working air gap magnetic field, S is the circumferential surface area of the coil, $S_{p1}$ and $S_{p2}$ are the circumferential surface area of the inner and outer permanent magnets, and $\mu_{0},\mu_{r}$ are the permeability of air and the relative permeability of soft magnets. In summary, the optimal magnetic induction intensity can be obtained by designing the parameters of the permanent magnet to meet the uniform distribution of magnetic induction intensity at the working air gap.

### 3.2. RLFMB Rotor Dynamics Model

After being activated, the coil in rotating Lorentz force magnetic bearings is exposed to the Lorentz force in the magnetic field. The coil travels through two working air gaps in series between the inner and outer magnetic steel and is subjected to two equal-sized and opposite-direction Lorentz forces that produce torque. When a torque is generated, the coil bracket’s output shaft is driven to rotate, achieving fixed shaft rotation. The coil bracket and output shaft both use mechanical transmission structures. This method is described as follows:(4)$$\left\{\begin{array}{l} F=\frac{H_{c}(2l_{p1}+2l_{p2})}{(\frac{l_{1}}{S_{1}\mu_{0}\mu_{r}}+\frac{l_{2}}{S_{2}\mu_{0}\mu_{r}}+\frac{2\delta }{\mu_{0}S}+\frac{2l_{p1}}{S_{p1}\mu_{0}\mu_{r}}+\frac{2l_{p2}}{S_{p2}\mu_{0}\mu_{r}})S}L_{a}i\\ T=\frac{H_{c}(2l_{p1}+2l_{p2})}{(\frac{l_{1}}{S_{1}\mu_{0}\mu_{r}}+\frac{l_{2}}{S_{2}\mu_{0}\mu_{r}}+\frac{2\delta }{\mu_{0}S}+\frac{2l_{p1}}{S_{p1}\mu_{0}\mu_{r}}+\frac{2l_{p2}}{S_{p2}\mu_{0}\mu_{r}})S}L_{a}Di\\ T=J\alpha +K_{f}\omega +K_{e}\theta \end{array}\right.$$

In the above formula, J is the moment of inertia of rotating Lorentz force bearings’ rotor assembly and load, $K_{f}$ is the frictional moment coefficient, $K_{e}$ is the elastic moment coefficient, α is the angular acceleration of the rotor, and ω is the angular velocity of the rotor.

The main part of the circuit of RLFMB is a coil, which is expressed by the following formula:(5)$$\left\{\begin{array}{l} u=Ri+L\frac{di}{dt}+e\\ e=BL_{a}\frac{d\theta }{dt} \end{array}\right.$$

In the above formula, u is the operating voltage, R is the equivalent resistance of RLFMB, and L is the equivalent inductance of RLFMB, that is, the inductance of the coil. e is the back EMF, generated by the motion of the coil cutting the magnetic inductive lines in a magnetic field. $L_{a}$ is the working length of the coil, θ is the angular displacement of the coil in RLFMB, B is the magnetic induction intensity, and i is the operating current in the equivalent circuit.

Simultaneously, (4) and (5) derive the transfer function from the input voltage to the rotor displacement:(6)$$\frac{\theta }{u}=\frac{BL_{a}D}{(JR+Lk_{f})s^{3}+(k_{f}R+k_{e}R+B^{2}L_{a}{}^{2}D)s^{2}+k_{e}Ls}$$

This can be rewritten into state-space form:(7)$$\begin{array}{l} \dot{x}=Ax+Bu\\ y=Cx \end{array}$$

In the above formula, x is the rotor state vector, u is the control input, and y is the rotor output. And
$$A=\left[\begin{array}{ccc} 0 & 1 & 0\\ 0 & 0 & 1\\ \frac{-k_{e}}{J} & \frac{-(k_{f}R+k_{e}R+B^{2}L_{a}{}^{2}D)}{LJ} & \frac{-(JR+Lk_{f})}{LJ} \end{array}\right]$$
$$B=\left[\begin{array}{c} 0\\ 0\\ 1 \end{array}\right]$$
$$C=\left[\begin{array}{ccc} \frac{BL_{a}D}{LJ} & 0 & 0 \end{array}\right]$$

Consider that when the stable platform is working, it is affected by the environmental torque, and the attitude actuator such as the thruster and flywheel excites the spacecraft and stabilizes the platform to generate vibration when working, which adds unknown disturbance term $f(x_{1},x_{2})$. Take the RLFMB rotor displacement and angular velocity as the controlled quantity at the same time, so that the vector
(8)$$x_{1}=\left[\begin{array}{c} \theta \\ \omega \end{array}\right]$$

The following is derived:$$\begin{array}{l} {\dot{x}}_{1}=x_{2}\\ {\dot{x}}_{2}=bu+f(x_{1},x_{2}) \end{array}$$

In the above formula, u is the RLFMB output.

RLFMB is a third-order system, as shown by Equation (6), and it can be controlled in accordance with the controllability condition. When designing the RLFMB structure and choosing magnetic steel, the best course of action can be planned to make the magnetic field at the air gap of the magnet uniform and to have a large phase margin and amplitude margin at the same time because each parameter is an actual physical quantity combined with Formula (3).

## 4. RLFMB Online Controller Design

### 4.1. Controller Performance Requirements Analysis

The following three requirements serve as the foundation for the design of the RLFMB controller:(1)The RLFMB’s internal and exterior magnets are not designed to close, and the rotor swings back and forth within the operating air gap. To prevent rotor and stator collisions, we design the RLFMB’s angular displacement output equation such that the angular velocity at the end point is 0.(2)RLFMB is oriented steadily to prevent needless shaking of the load compartment and frame compartment. We created an adaptive controller based on an RBF network, and we minimized input signal overshoot and static error. Additionally, the impact of any random system disturbances should be swiftly minimized.(3)We introduced a current feedback inner loop to improve the system’s response time and control bandwidth while preventing excitation vibration of the input signal.

The current widely used technique is to provide the input equation and construct the actuator as a program control system in order to limit the speed and position of the actuator at a specific point or over a specific period of time.

The relevant rotor motion equation is determined from the input signal.

For RLFMB, there are
(9)$$Bi(t)L_{a}D=J\alpha ,$$

The angular acceleration is expressed as
(10)$$\alpha (t)=\frac{Bi(t)L_{a}D}{J}=ki(t),$$

In the above formula, k is the angular acceleration coefficient. It can be seen that the control of the current signal can realize the control of the diagonal acceleration. This paper uses the sinusoidal function as the rotor motion equation for RLFMB:(11)$$\theta (t)=\frac{A\sin\omega_{e}t}{\omega_{e}k},$$

After seeking guidance, there is
(12)$$\omega (t)=\frac{A\sin\omega_{e}t}{k},$$

Let $\omega_{t}$ be the ordinate and θ(t) be the abscissa, so there is
(13)$${\left(\frac{\omega (t)k}{A}\right)}^{2}+{\left(\frac{\theta (t)\omega_{e}k}{A}\right)}^{2}=1.$$

This is an elliptical line’s upper half. When using the highest value, it comes to zero. As a result, the angular velocity is zero when the rotor is at the magnet’s edge, preventing it from touching the stator.

### 4.2. Design of Adaptive Control Method Based on RBF Network

The tracking error can be employed as an input index and quickly converges to zero in a finite amount of time if RLFMB is required to have a modest static difference and overshoot of the input signal.

As a single-input, single-output system, think of RLFMB:(14)$$\begin{array}{l} {\dot{x}}_{1}=x_{2}\\ {\dot{x}}_{2}=bu+f(x_{1},x_{2}) \end{array}.$$

In the above formula, u is the control input, while $x_{1},x_{2}$ are the angular displacement signal and angular velocity signal, respectively. Because the controller input is not constant at zero, take $u=\sigma u_{c}$, where 0<σ<1. If the angle position input is $x_{d}$ and the tracking error is $e=x_{1}-x_{d}$, then $\overset{\cdot }{e}=x_{2}-{\overset{\cdot }{x}}_{d}$.Take position instruction $x_{d}=\frac{A\sin\omega_{e}t}{\omega_{e}k}$ and design the control law when t→∞, $e\to 0,\overset{\cdot }{e}\to 0$.

Equation (5) shows that there is an induced electromotive force disturbance and that the input voltage of the RLFMB is not linear with the coil current. The spacecraft is also affected by environmental torque, and the stable platform is affected by other attitude actuators, such as the flywheel and thruster, which will result in an increase in pointing error or even the target being lost during the attitude maneuver.

Let $f(x_{1},x_{2})$ be a function of angular displacement and angular velocity; the specific parameters and form are unknown, and we simulate the disturbance during the stable pointing of RLFMB.

The approximation of the unknown perturbation function $f(x_{1},x_{2})$ can be implemented using the RBF network. The network algorithm is
(15)$$\begin{array}{l} h_{i}=g({\left\|x-c_{ij}\right\|}^{2}/b_{j}{}^{2})\\ f=W^{*T}h(x)+\varepsilon \end{array}.$$

In the above formula, x is the network input, i is the number of inputs to the network, j is the jth node of the implied layer, $h={\left[h_{1},h_{2},\ldots h_{n}\right]}^{T}$ is the output of a Gaussian function, $W^{*}$ is the ideal weight for the network, ε is the error approximation of the network, and $\left|\varepsilon \right|\leq \varepsilon_{N}$.

Using RBF to approximate the unknown function $f(x_{1},x_{2})$, the input of the network takes $x={\left[x_{1}\;\;x_{2}\right]}^{T}$, and then the output of the RBF network is
(16)$$\overset{\wedge }{f}(x)=\overset{\wedge }{W^{T}}h(x),$$

Further expansion yields
(17)$$\tilde{f}(x)=f(x)-\overset{\wedge }{f}(x)=W^{*T}h(x)+\varepsilon -\overset{\wedge }{W^{T}}h(x)=\tilde{W^{T}}h(x)+\varepsilon ,$$
and we define $\tilde{W}=W^{*}-\overset{\wedge }{W}$.

The designed slipmode function is
(18)$$s=ce+\overset{\cdot }{e},$$

In the above formula, c>0, and
(19)$$\overset{\cdot }{s}=c\overset{\cdot }{e}+\overset{\cdot \cdot }{e}=c\overset{\cdot }{e}+b\sigma u_{c}+f(x)-\overset{\cdot \cdot }{x_{d}}=c\overset{\cdot }{e}+\beta u_{c}+f(x)-\overset{\cdot \cdot }{x_{d}},$$

In the above formula, β=bσ. Taking $\lambda =\frac{1}{\beta }$, we designed the Lyapunov function as
(20)$$V=\frac{1}{2}s^{2}+\frac{\left|\beta \right|}{2\gamma }{\tilde{\lambda }}^{2}+\frac{1}{2\gamma }\tilde{W^{T}}\tilde{W}.$$

In the above formula, $\tilde{\lambda }=\overset{\wedge }{\lambda }-\lambda ,\gamma >0$. The following is a derivation of Equation (20):(21)$$\overset{\cdot }{V}=s\overset{\cdot }{s}+\frac{\left|\beta \right|}{\gamma }\tilde{\lambda }\overset{\cdot }{\tilde{\lambda }}-\frac{1}{\gamma }\tilde{W^{T}}\overset{\cdot }{\overset{\wedge }{W}}=s(c\overset{\cdot }{e}+\beta u_{c}+f(x)-\overset{\cdot \cdot }{x_{d}})+\frac{\left|\beta \right|}{\gamma }\tilde{\lambda }\overset{\cdot }{\overset{\wedge }{\lambda }}-\frac{1}{\gamma }\tilde{W^{T}}\overset{\cdot }{\overset{\wedge }{W}},$$

Take
(22)$$\begin{array}{l} \alpha =ks+c\overset{\cdot }{e}+\overset{\wedge }{f}(x)-\overset{\cdot \cdot }{x_{d}}+\eta \mathrm{sgn}s\\ k>0,\eta \geq \varepsilon_{N} \end{array},$$

Equation (20) can be further expanded to
(23)$$\begin{array}{l} \overset{\cdot }{V}=s(\alpha -ks+\beta u_{c}+\tilde{f}(x)-\eta \mathrm{sgn}s)+\frac{\left|\beta \right|}{\gamma }\tilde{\lambda }\overset{\cdot }{\overset{\wedge }{\lambda }}-\frac{1}{\gamma }\tilde{W^{T}}\overset{\cdot }{\overset{\wedge }{W}}=\\ s(\alpha -ks+\beta u_{c}+\varepsilon -\eta \mathrm{sgn}s)+\frac{\left|\beta \right|}{\gamma }\tilde{\lambda }\overset{\cdot }{\overset{\wedge }{\lambda }}+\tilde{W^{T}}(sh(x)-\frac{1}{\gamma }\overset{\overset{\cdot }{\wedge }}{W}) \end{array},$$

The design control law and adaptive law are
(24)$$\begin{array}{l} u_{c}=-\hat{\lambda }\alpha \\ \dot{\hat{\lambda }}=\gamma s\alpha \mathrm{sgn}b\\ \dot{\hat{W}}=\gamma sh(x) \end{array}.$$

And
(25)$$\begin{array}{l} \overset{\cdot }{V}=s(\alpha -ks-\beta \overset{\wedge }{\lambda }\alpha +\varepsilon -\eta \mathrm{sgn}s)+\frac{\left|\beta \right|}{\gamma }\tilde{\lambda }\gamma s\alpha \mathrm{sgn}\beta \\ =s(\alpha -ks-\beta \overset{\wedge }{\lambda \alpha })+\beta s\alpha \tilde{\lambda }=s(\alpha -ks-\beta \alpha \lambda +\varepsilon )\\ \leq -ks^{2}\leq 0 \end{array},$$

Since $V\geq 0,\overset{\cdot }{V}\leq 0$, then V is bounded. When t→∞, s→0, and thus $e\to 0,\overset{\cdot }{e}\to 0$.

In conclusion, the function of the RBF network-based adaptive controller can be stated as follows: when the RLFMB is subjected to unknown interference, the angular displacement error and angular velocity of the rotor quickly converge to zero, ensuring that the rotor continues to rotate along the intended angular displacement output equation. Figure 7 is a block diagram of the adaptive control strategy.

In summary, the task of the adaptive controller based on the RBF network can be described as follows: when the RLFMB is subjected to unknown interference, the angular displacement error of the rotor e and angular velocity error $\overset{\cdot }{e}$ quickly converge to 0, ensuring that the rotor continues to rotate along the planned angular displacement output equation.

### 4.3. Controller Optimization Based on Current Feedback

The third-order SISO system RLFMB can be reduced to a normal second-order system by inferring from Equation (6) that its third-order term coefficient is relatively small. To achieve the suppression of the disturbance under the influence of the disturbance torque, the system rigidity must be raised and the system damping must be enhanced. In order to avoid the peak region that is prone to resonance, it is necessary to take into account both the specified angular displacement signal frequency and the undamped natural oscillation frequency of RLFMB while constructing the closed-loop control structure. It is specifically as follows.

For a typical second-order system, when the input signal is sinusoidal, there are
(26)$$\dot{\theta }(t)+2\xi \omega_{n}\dot{\theta }(t)+\omega_{n}{}^{2}\theta (t)=\frac{\omega_{n}{}^{2}A\sin(\omega_{e}t)}{\omega_{e}k},$$

Its amplitude-frequency characteristics are
(27)$$H(\omega_{e})=\frac{A/k\omega_{e}}{\sqrt{{\left(1-{\left(\omega_{e}/\omega_{n}\right)}^{2}\right)}^{2}+{\left(2\xi \omega_{e}/\omega_{n}\right)}^{2}}}.$$

$\omega_{n}$ is the undamped natural oscillation frequency of a typical second-order system, while ξ is the damping ratio. From Equation (27), it can be seen that the closer $\omega_{n}$ and $\omega_{e}$ are, the higher the excitation of the input signal to the system and the greater the vibration peak.

When RLFMB is operating, the mechanical electromagnetic parameters should be reasonably constructed so that the system’s natural oscillation frequency avoids the frequency of the input signal in order to prevent excessive vibration excitation generated by the output signal to the system. In order to give the RLFMB closed-loop control system an inner loop with a faster response and wider bandwidth, more feedback channels can be added in addition to angular displacement and angular velocity feedback. This will help to address the shortcomings of narrow angular displacement and angular velocity feedback bandwidth.

It can be seen from Equation (24) that the purpose of the velocity and displacement closed loop is to converge the velocity and displacement deviation to zero in a limited time, while at the same time improving the rapidity of the system. The three-ring control structure is shown in Figure 8.

The current loop can be expressed as follows:(28)$$\left\{\begin{array}{l} T_{y}=BiL_{a}D=J\ddot{\theta }\\ i=\frac{J\ddot{\theta }}{BL_{a}D}=-g(s)A\\ g(s)=K_{p}+\frac{K_{i}}{s}+K_{d}s \end{array}\right..$$

In the above formula, i is the control current of RLFMB, g(s) indicates the current control rate of RLFMB, and A is the measured value of the sensor. The controller for current feedback uses traditional PID control because the purpose of current feedback is to ensure that the actual current of the coil closely follows the reference current, thus ensuring the implementation of the given angular displacement equation. PID control is simple and easy to implement and can be placed directly on the RLFMB’s drive circuit.

## 5. Simulation Results and Analysis

RLFMB is a servo mechanism for a Lorentz force stabilized platform, and aerospace servo mechanisms are often controlled using PID control, robust sliding mode control, etc. In the simulation analysis presented herein, the adaptive controller created in this study is compared to PID and robust sliding mode controllers. The tracking performance of the supplied signal, the disturbance immunity performance of step disturbance signal, and continuous disturbance signal are first examined before comparing the diagonal velocity tracking performances of the adaptive controller and robust sliding mode controller.

### 5.1. Angular Displacement Signal Tracking Error Comparison

The traditional PID controller is straightforward and simple to use, but it has a lot of empirical components and complex percentage, integral, and differential coefficients to change. The PID settings are optimized using a genetic algorithm. The absolute value of the error time integral performance index is employed as the minimal objective function of parameter selection in order to achieve satisfactory dynamic features of the transition process. The penalty function is used, and the overshoot is used as the term of the ideal index in order to minimize the overshoot as much as is feasible.

In our work, the crossover probability was set to 0.9, the mutation probability to 0.03, and the initial population size was set at 30. The proportional coefficient was 9.7528, the integral coefficient was 2.2521, and the differential coefficient was 0.0817 after iteratively optimizing PID parameters.

Because the robust sliding mode controller needs to define the upper and lower bounds of the disturbance, the upper bound is 10% of the input signal, and the lower bound is 0. In the approach law, the coefficient of the sign function ε is 0.1 and the coefficient of the power function k is 1.

Table 1 shows the simulation parameters of the adaptive controller. Table 2 gives the structural parameters of RLFMB, and the parameters of Table 2 are brought into Equation (26). The undamped natural oscillation frequency of the system is 0.0768, and when the input sinusoidal signal meets $\omega_{e}>>\omega_{n}$, the amplitude of the excited vibration is approximately equal to 0.

In summary, let the given angular displacement signal be θ(t)=sint. In the legend of the simulation diagram, Adaptive refers to the adaptive controller, PID refers to the PID controller, and Robust refers to the robust sliding mode controller. The simulation results are shown in Figure 9.

As can be seen in Figure 9, the output of the adaptive controller, for a given sinusoidal signal, almost never overshoots and can converge the tracking error to the minimum the first time, whereas the static error of the PID controller and the robust sliding mode controller cannot be eliminated even after many oscillations. As a result, the RLFMB can quickly cause the load compartment to rotate at a specified angle and prevent repeated swing thanks to the adaptive controller. This not only enhances the satellite load’s capacity to maneuver with agility but also prevents the recurrent oscillations that cause other components to vibrate.

### 5.2. Comparison of Rejection Performance of Perturbed Signals

A pulse disturbance with an amplitude of 1 rad is applied to the adaptive controller, PID controller, and robust sliding mode controller, respectively, at 20 s. 

The simulation results are shown in Figure 10 and Figure 11.

The adaptive controller has the smallest response to the pulse signal, the fastest convergence speed, and no repetitive oscillation, as can be seen in Figure 10. The convergence amplitude is excessively large, which makes it possible to damage the actuator and cause high-frequency oscillation, even if the strong sliding mode controller can quickly reduce the influence of pulse disruption. In order to successfully avoid the system-impacting effects of rapid signals like pulse and step, RLFMB forces the load compartment to spin. The adaptive controller then swiftly converges the pointing error to ensure the stability of the load pointing.

The RLFMB receives an application of a multiple-frequency sinusoidal signal with a maximum frequency of 10 Hz and an amplitude of 0.1 rad. The simulation results are shown in Figure 11.

As can be seen in Figure 11, an adaptive controller can maintain good directional stability even in the presence of a constant disturbance signal. The errors are fewer and the curves are smoother than with PID controllers and robust sliding mode controllers, even though some mistakes cannot be completely removed. The impact of the disturbance signal is negligible and the pointing precision is higher when the RLFMB rotates the load chamber.

### 5.3. Tracking Error for Diagonal Velocity

The adaptive controller also safeguards the rotor from collisions by tracking diagonal velocity to ensure that the RLFMB has an endpoint velocity of zero.

As can be seen in Figure 12, the adaptive controller can nearly instantly follow the angular velocity of the RLFMB rotor. The adaptive controller has a smaller overrun when a specific signal has just been applied, guaranteeing that the RLFMB drives the load compartment more smoothly. After stabilization, the adaptive controller’s diagonal velocity tracking may achieve zero error, guarantee that the angular velocity of the RLFMB driving the load chamber is always under control, and prevent load chamber instability brought on by the rotor of the RLFMB striking the stator.

## 6. Conclusions

In order to demonstrate the significance of spinning magnetic bearings for achieving stable pointing of loads, this work first offered a novel seven-degree-of-freedom Lorentz force stabilized platform. Next, it simulated the dynamics of rotating magnetic bearings. Analyzing the structural characteristics and functionality of the spinning magnetic bearing allowed for the creation of an adaptive controller based on the RBF network. The experiment’s results showed how effectively the adaptive controller tracks angular velocity signals, suppresses disturbance signals, and tracks angular displacement signals. This study is crucial for improving the pointing stability and precision of the satellite/payload since it offers solutions for related problems in the realms of laser communication and high-resolution reconnaissance.

## Figures and Tables

**Figure 1 sensors-23-08543-f001:**
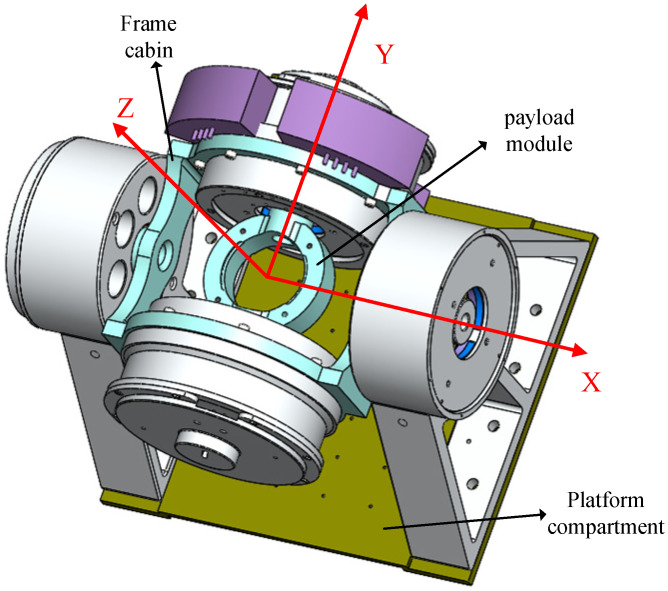
RLFMB control load compartment/frame compartment deflection.

**Figure 2 sensors-23-08543-f002:**
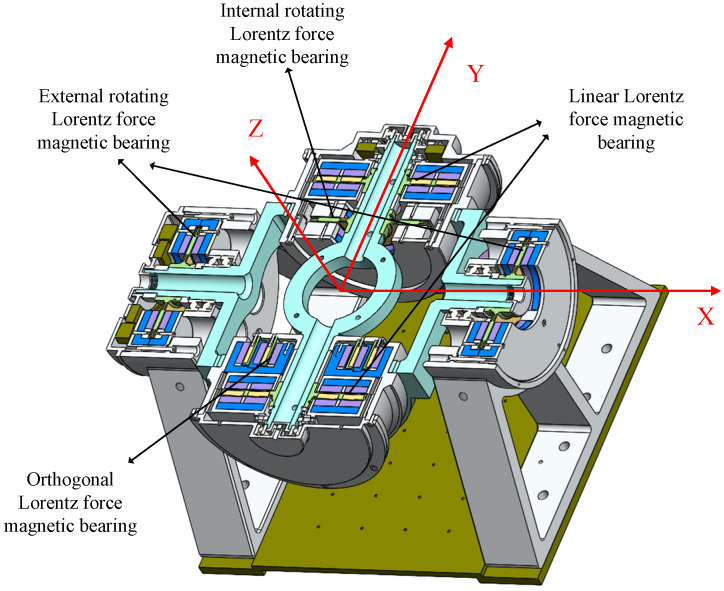
Schematic diagram of the overall configuration of the universal stable platform.

**Figure 3 sensors-23-08543-f003:**
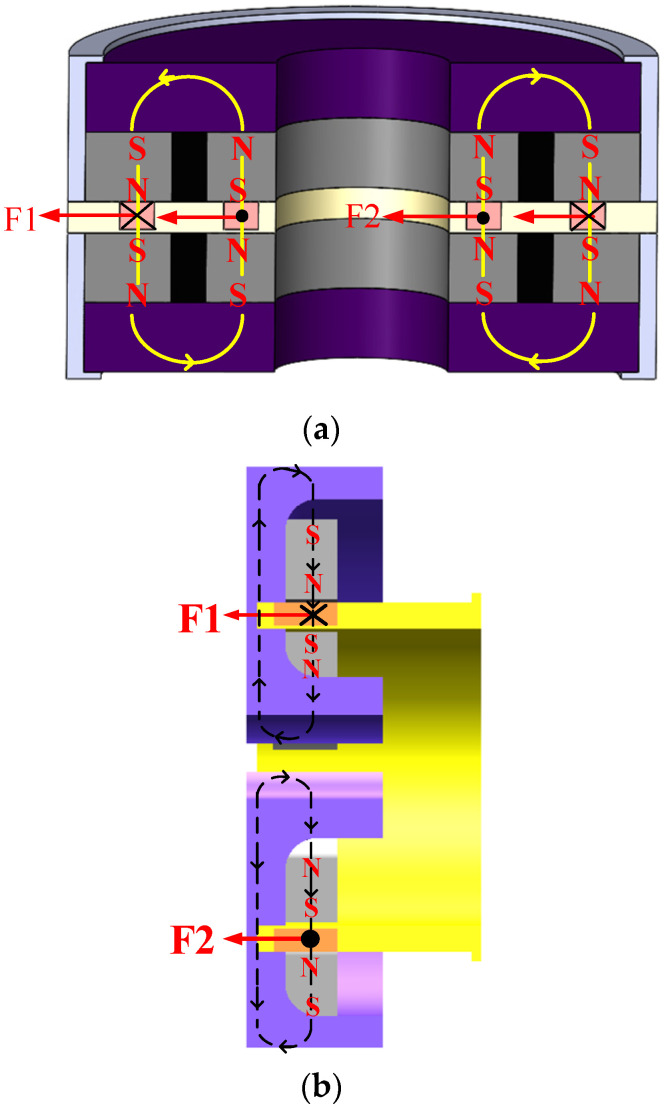
Force diagram of orthogonal, linear Lorentz force magnetic bearings. (**a**) Force diagram of orthogonal magnetic bearings. (**b**) Force diagram of linear magnetic bearings.

**Figure 4 sensors-23-08543-f004:**
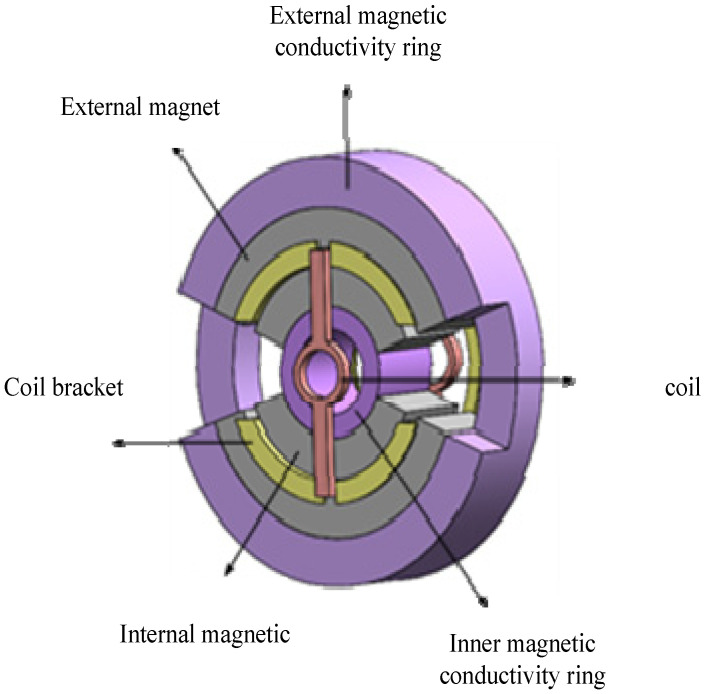
RLFMB structure.

**Figure 5 sensors-23-08543-f005:**
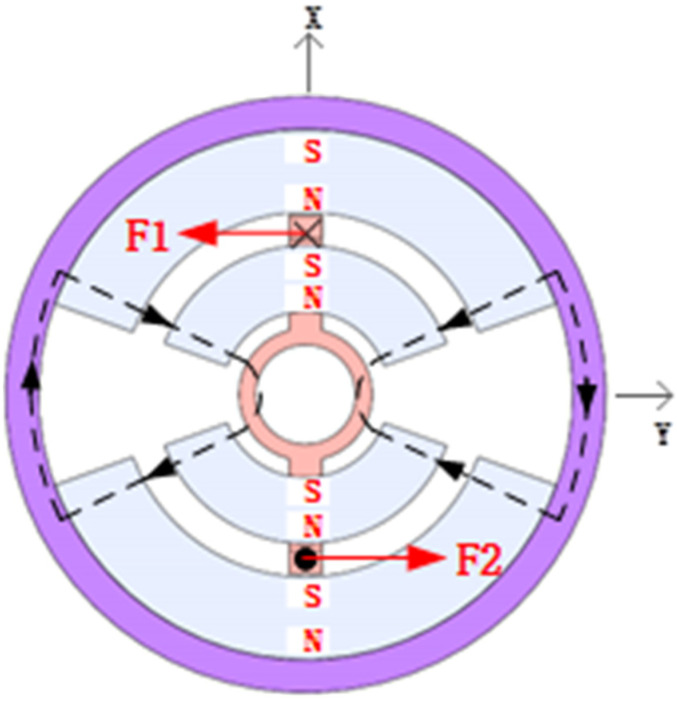
RLFMB output torque.

**Figure 6 sensors-23-08543-f006:**
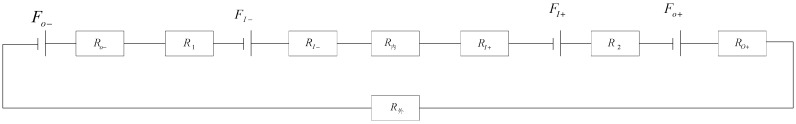
RLFMB equivalent magnetic circuit.

**Figure 7 sensors-23-08543-f007:**
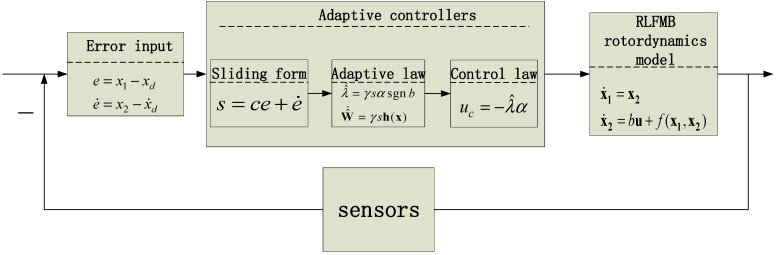
Structure of adaptive fault-tolerant control strategy.

**Figure 8 sensors-23-08543-f008:**
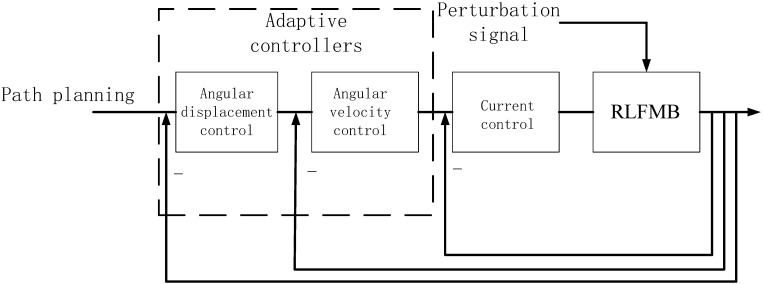
Structure of RLFMB three-loop control.

**Figure 9 sensors-23-08543-f009:**
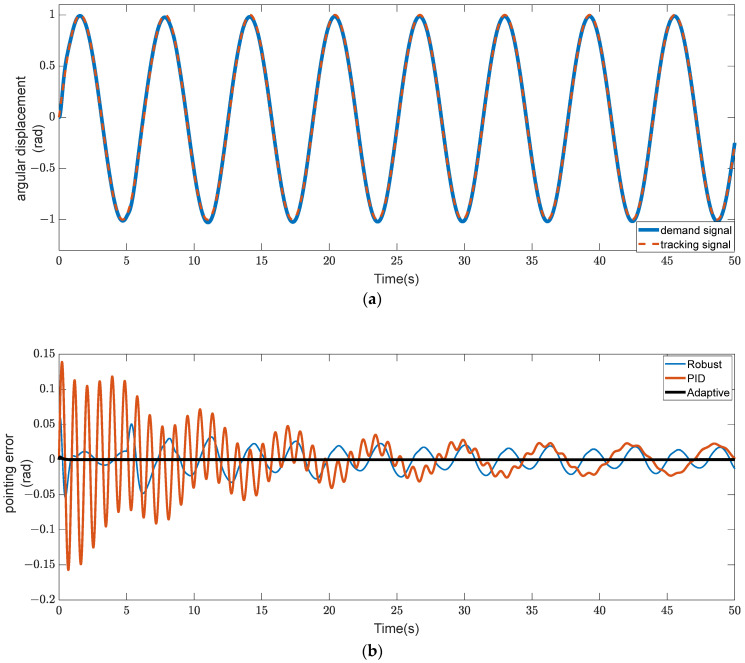
Comparison of angular displacement signal pointing error. (**a**) Angular displacement tracking chart. (**b**) Angular displacement pointing error curve.

**Figure 10 sensors-23-08543-f010:**
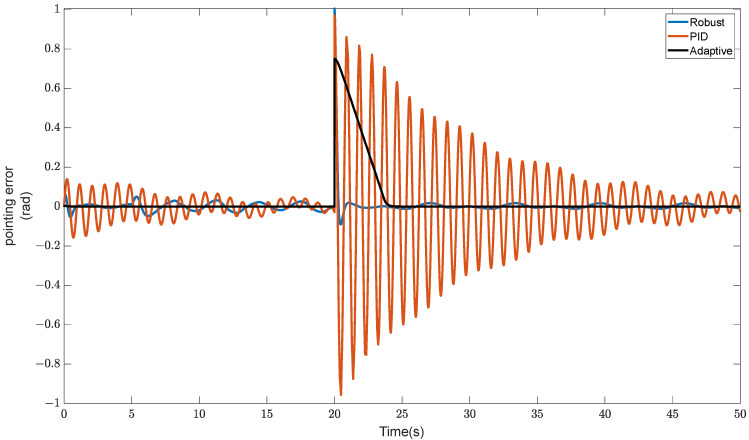
Comparison of anti-interference performance under pulse signals.

**Figure 11 sensors-23-08543-f011:**
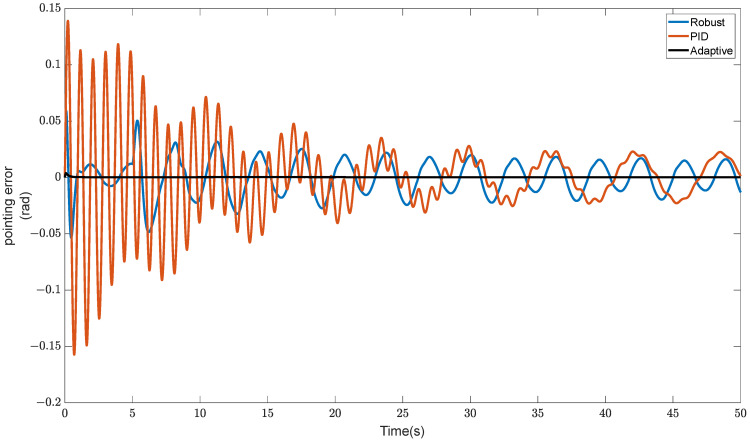
Comparison of anti-interference performance under pulse signals.

**Figure 12 sensors-23-08543-f012:**
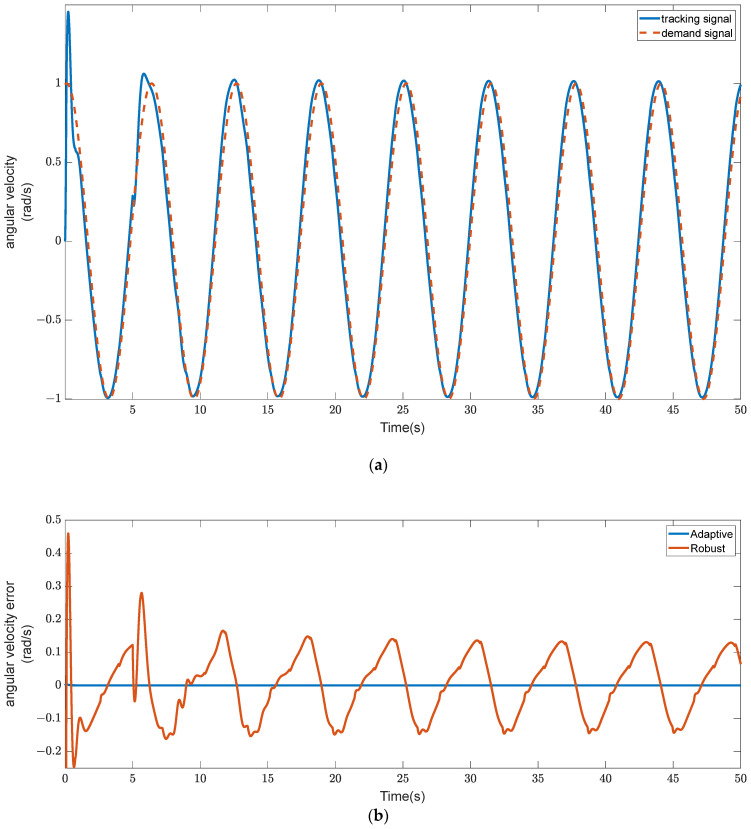
Angular velocity tracking performance. (**a**) Angular velocity tracking chart. (**b**) Error curve.

**Table 1 sensors-23-08543-t001:** Controller simulation parameters.

Items	Value
k	5
γ	10
η	0.1
$c_{i}$	[–2, –1, 0, 1, 2]
$b_{i}$	3.0
Initial value of the grid weight element	0.1
$\overset{\wedge }{\lambda }(0)$	1.0
ε	0.1
k	1

**Table 2 sensors-23-08543-t002:** RLFMB structure parameters.

Items	Value
B/(T)	0.5
$L_{a}$/(m)	4
D/(m)	0.1
L/(H)	0.1
R /(Ω)	3.0
J$/(\mathrm{kg}\cdot m^{2}$)	0.425

## Data Availability

In this study, no new data were created.

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
