# Peer review of "Rotating Lorentz Force Magnetic Bearings’ Dynamics Modeling and Adaptive Controller Design"

_sensors, 2023, doi:10.3390/s23208543_

Round 1

Reviewer 1 Report

Report on the manuscript

Title:  Rotating Lorentz force magnetic bearing dynamics modeling and adaptive controller design

Authors: Feiyu Chen, Shengjun Wang and Weijie Wang

Manuscript ID: sensors-2599115

In the paper the authors studied the dynamic modeling and control method of Rotating Lorentz force magnetic bearing (RLFMB). I think the readers of this journal will appreciate the results of this interesting manuscript.  We consider that the manuscript is well written, the material is judiciously divided and organized and correct from scientific point of view. Some changes are, however, necessary. For these reasons I can recommend the acceptance of this paper after some corrections.

Before that the Editor makes a decision, I suggest that the authors emphasize take into account the following corrections

1.    The Abstract section is presented too superficially and without pointing out what is really being done. Please improve this section.

2.    Please highlight how the work advances or increments the field from the present state of knowledge and provide a clear justification for your work.

3.    The section Conclusions will be point out the original results of the paper and can be extended to highlight the contributions. Please provide a clear justification for your work in this section, and indicate uses and extensions if appropriate.

4.    The conclusion section has to be rewritten doing an effort to remark the main findings rather than summarizing the article content.

5.    Please check the paper again for any possible misprints.

6.    The text needs to be checked and revised by a native speaker or a language expert. You may consider (at your own cost) the use of a possible professional copyediting service

7.    After each relationship a point, comma or semi-column should be placed.

8.    I think the authors need to emphasize more clearly the contribution of the manuscript from a scientific point of view.

9.    Template of the journal must be respected.

10.  The quality of the Figures must be improved.

 If the author takes into account these observations the work can be published.

Author Response

Dear Professor,

Thank you very much for your valuable suggestions. I have revised the paper in response to your suggestions. The following is a reply to your suggestion:

  1. The abstract has been revised and the innovation of this paper has been pointed out in the abstract;
  2. The importance and necessity of innovation points in this paper are laid out according to the existing technology;
  3. Rewrote the conclusion, clarified the reason for the work and made appropriate expansionï¼›
  4. Perfect English, check spelling and grammar errorsï¼›
  5. Improved formula format and article formatï¼›
  6. Improved the simulation experiment, added a new comparison group, and scientifically emphasized the contribution of this paper.

Reviewer 2 Report

The introduction provide sufficient background and include many relevant references but these references aren't combined with the topic of the article. There is a lack of detail on the methods used.The figure and Table in the experiment section need to be checked carefully.

The quality of english language need moderate editing.

Author Response

Dear Professor,

Thank you for your valuable comments. I have revised the article according to your comments, and improved the abstract and introduction, so that the references are closely integrated with the topic of the article. In this paper, the innovation points are pointed out, the simulation experiment is improved, and a new comparison group is added.

Reviewer 3 Report

Dear Authors,

In my opinion, the research presented in the paper is relevant. Nevertheless, I have a few remarks:

1. The English language requires significant improvement.

2. Terminology is incorrect, for example: external magnetic conductivity ring, leakage magnetism, magnetization length, etc.

3. In Tab. 2 please change words to English

The English language requires significant improvement, especially used terminology is incorrect.

Author Response

Dear Professor,
Thank you for your valuable comments! I have revised the paper according to your suggestions, strengthened the English expression of the paper, improved the terminology, and replaced Table 2 with English.

Reviewer 4 Report

The introduction of the article is too large but no quality information is providing.

It is suggested to write why the proposed method is superior than other?

It would be better to add comparative analysis to provide the necessity of proposed technique.

Fig 2 is looks like completely existing one, whats the new innovation in it.

The methodology of the article is not clear

Article contains more theoretical data rather than results.

The presented results are not sufficient to prove the concept and excellency of the work.

It is suggested to add more comparative results with existing methods, 

Due to these major limitations article cant be proceed further in its present stage.

The introduction of the article is too large but no quality information is providing.

It is suggested to write why the proposed method is superior than other?

It would be better to add comparative analysis to provide the necessity of proposed technique.

Fig 2 is looks like completely existing one, whats the new innovation in it.

The methodology of the article is not clear

Article contains more theoretical data rather than results.

The presented results are not sufficient to prove the concept and excellency of the work.

It is suggested to add more comparative results with existing methods, 

Due to these major limitations article cant be proceed further in its present stage.

Author Response

Dear Professor,
Thank you for your valuable comments! I have revised the article according to your comments, and I will respond to your questions below:
1. The innovation points of this paper are clarified, and the importance and innovation of the 7-DOF Lorentz force magnetic levitation stabilization platform are stated in the introduction and the main body;
2. Improved the drawings and tables, replacing the concept drawings with engineering drawings for Figure 2. The new structure, shown in Figure 2, can isolate the vibration of the spacecraft to the satellite/payload and actively drive the direction of the payload. Different from the traditional stable platform, the 7-DOF platform in this paper is completely supported by the Lorentz force magnetic bearing, with linear output, high control accuracy, and no negative current stiffness, avoiding the error caused by the output nonlinear of the reluctance magnetic bearing. There are no mechanical bearings, and 7 degrees of freedom can be moved, can achieve a wider range of maneuvering and stable pointing. The detailed discussion is reflected in the introduction of this paper.
3. According to your valuable comments, a new comparison group was added to the simulation experiment to demonstrate the superiority of the method in this paper.

Round 2

Reviewer 3 Report

Dear Authors,

thank you for taking into account all my remarks.

Reviewer 4 Report

Article has been improved, now article in a good shape to process furhter.

Article has been improved, now article in a good shape to process furhter.